**www.cambridge.org/ext**

Early Stone Age; Oldowan; temporal cohesion; cultural loss; convergent evolution

**Corresponding author:**
Alastair Key;
Email: ak2389@cam.ac.uk

# A continuous record of early human stone tool production

Alastair Key ⬤ and Eleanor M. Williams

Department of Archaeology, University of Cambridge, UK

## Abstract

Early human cultural dynamics underpin the Plio-Pleistocene archaeological record and impact how we understand some of our earliest identifiable behaviours. One major outstanding question is whether Early Stone Age material culture represents a single lineage of cultural information, or did we ever lose the knowledge required to make stone tools? No single approach satisfactorily addresses this problem, but to date, objective analyses of temporal data have been absent from the conversation. Here, using a comprehensive database of dated African Oldowan archaeological sites, we demonstrate that there are no temporal breaks large enough, on a relative basis, to infer a loss of stone-tool-making cultural information. Therefore, alongside previously published data, we infer a continuous record of early human stone tool production in Africa from c. 3.3 to 1.5 million years ago. Stone tool-associated behavioural adaptations and evolutionary selective pressures were, therefore, likely to have been ever present during this period.

## Impact statement

Flaked stone technologies revolutionised the hominin adaptive niche and provided significant selective pressures on human cognitive and anatomical evolution. We address three major questions regarding early human stone tools: Was their use, benefit and evolutionary influence constant before 1.5 million years ago (Ma)? Moreover, did we ever forget how to make stone tools, and can the Oldowan be considered a cohesive cultural tradition? Using a comprehensive sample of African Oldowan sites and frequentist statistical models, we demonstrate that there is no temporal evidence for a loss of stone-tool-making knowledge 3.3–1.5 Ma. Stone tools appear to have constantly benefited hominins during this period and provided an ever-present adaptive role, reinforcing their importance to the human story.

## Introduction

Early Stone Age (ESA) cultural dynamics are (Lycett, 2013; Toth and Schick, 2018; Stout et al., 2019), and always have been (Leakey, 1971; Isaac, 1984), relatively poorly understood, yet they underpin the Plio-Pleistocene archaeological record and impact how we understand early human behaviour. Hampered by a sparse and coarsely dated artefact record limited almost entirely to stone tools (Isaac, 1972; Schick and Toth, 2006; Key and Proffitt, 2024; Finestone, 2025), archaeologists rely on technological and morphological similarities between temporally heterogeneous occurrences (de la Torre et al., 2003; Stout et al., 2010; Braun et al., 2019; Delagnes et al., 2023), or data derived from extant referents (Carvalho and McGrew, 2012; Stout et al., 2019; Eren et al., 2020; Bandini et al., 2022; Snyder et al., 2022; Clark, 2025), to infer cultural and behavioural links between populations. Neither approach satisfactorily addresses one of the most important outstanding questions concerning our earliest material culture: does it represent a single, braided lineage of cultural information passed on through generations over millions of years, or did we ever lose the knowledge required to make stone tools?

The Oldowan represents the earliest widespread human material culture (Toth and Schick, 2018; Plummer et al., 2023; Finestone, 2025) and the best candidate for identifying a potential episode of ESA cultural loss. Produced for 1.6–2.0 million years by (likely) more than one species of hominin with cognition, anatomy and diets mosaically (*c.f.*, Kivell et al., 2023) adapted to retaining flaked stone tool material culture (Marzke, 2013; Antón et al., 2014; Shea, 2017; Lüdecke et al., 2018; Patterson et al., 2019; Bruner and Beaudet, 2023; Kivell et al., 2023; Plummer et al., 2023; Braun et al., 2025; Williams et al., 2025), the potential to lose lithic cultural knowledge *could* have been ever present. Climatic/ecological changes impacting adaptive strategies, insufficient population sizes for complex material culture or increased predation pressure altering relevant cost:benefit ratios are a few of many possible scenarios leading to cultural loss.

If Oldowan cultural information was ever lost, and a similar culture later re-emerged, this phenomenon could archaeologically manifest as an exceptional temporal gap between occurrences, a stark shift in spatial presence or a notable change in technological attributes.

Here, following earlier investigations into the temporal-cohesion of the African Acheulean record (Key, 2022), and the Lomekwi 3 occurrence relative to Oldowan sites (Flicker and Key, 2023), we investigate the temporal cohesion of the Oldowan in Africa.

## Methods

Ninety one reliably dated Oldowan occurrences are currently known in Africa (as described by Williams et al. [2025], updated to include Namorotukunan [Braun et al., 2025]; Figure 1). The central tendency ages of these sites range from 2.90 to 1.47 million years ago (Ma), but their upper and lower age-range thresholds cover 3.44–1.26 Ma. Those younger than c. 1.6 Ma are arguably contentiously assigned and are not considered here, following Williams et al. (2025). Occurrences are currently known from South Africa, Kenya, Tanzania, Ethiopia and Algeria. Only four potential breaks in the Oldowan's temporal record can be visually observed (Figure 1). A c. 125,000-year-long gap exists between the earliest currently known Oldowan occurrence, Nyanaga (Kenya), and the next earliest at Namorotukunan-1 in Ethiopia (Table 1). A c. 170,000-year gap then exists between the central age estimate of Namorotukunan-1 and the third-earliest Oldowan site, Ledi-Geraru (Ethiopia). Approximately 90,000 years separate Ain Boucherit in Algeria and A.L. 666 in the Hadar region of Ethiopia (Table 1). Finally, a c. 180,000-year-long break exists between Sterkfontein Member 5 (South Africa) and Beds KS1–3 at Kanjera South (Kenya) (Table 1). These breaks are hereafter referred to as the 'Nyanyanga', 'Namorotukunan', 'Ain Boucherit' and 'Sterkfontein' temporal gaps, respectively. Otherwise, the Oldowan record is remarkably cohesive relative to the precision of current dating methods (Figure 1; Supplementary Information).

We applied Solow and Smith's (2005) surprise test to assess cohesion between an outlying temporal occurrence – in this case, the first or last Oldowan site before/after a temporal gap – and a larger sample of consecutive earlier or later occurrences, relative to the direction of the test. The method tests the null hypothesis that the outlying record '*was generated by the same process*' that created the earlier or later records (Roberts et al., 2023: 464). Simply put, is the outlying record exceptional relative to the temporal distribution observed in the larger sample of values? Described widely elsewhere (e.g., Solow et al., 2006; Kjeldsen and Prosdocimi, 2016; Roberts et al., 2021; Key, 2022), the surprise test assumes the larger sample ($k$), against which the outlier is tested, represents the largest or smallest values from a larger distribution from the Gumbel domain of attraction. We refer the reader to these earlier studies for other model assumptions. Use of the Gumbel distribution is appropriate in light of the central ages displayed in the Oldowan sample (Table 1; Supplementary Information). Note that the date data fit a Weibull distribution, which in itself supports the presence of a continuous Oldowan record (Supplementary Information). First formulaically expressed in human origins research by Roberts et al. (2023), and copied here, Solow and Smith (2005) demonstrated the quantity,

$$S_k = \frac{y - t_1}{(y - t_1) + \Sigma_{j=1}^{k-1}(j+1)(t_j - t_{j+1})},$$

has a β distribution with parameters 1 and $k$-1, so that the *P*-value corresponding to an observed value $S_k$ is

$$P = (1 - S_k)^{k-1}$$

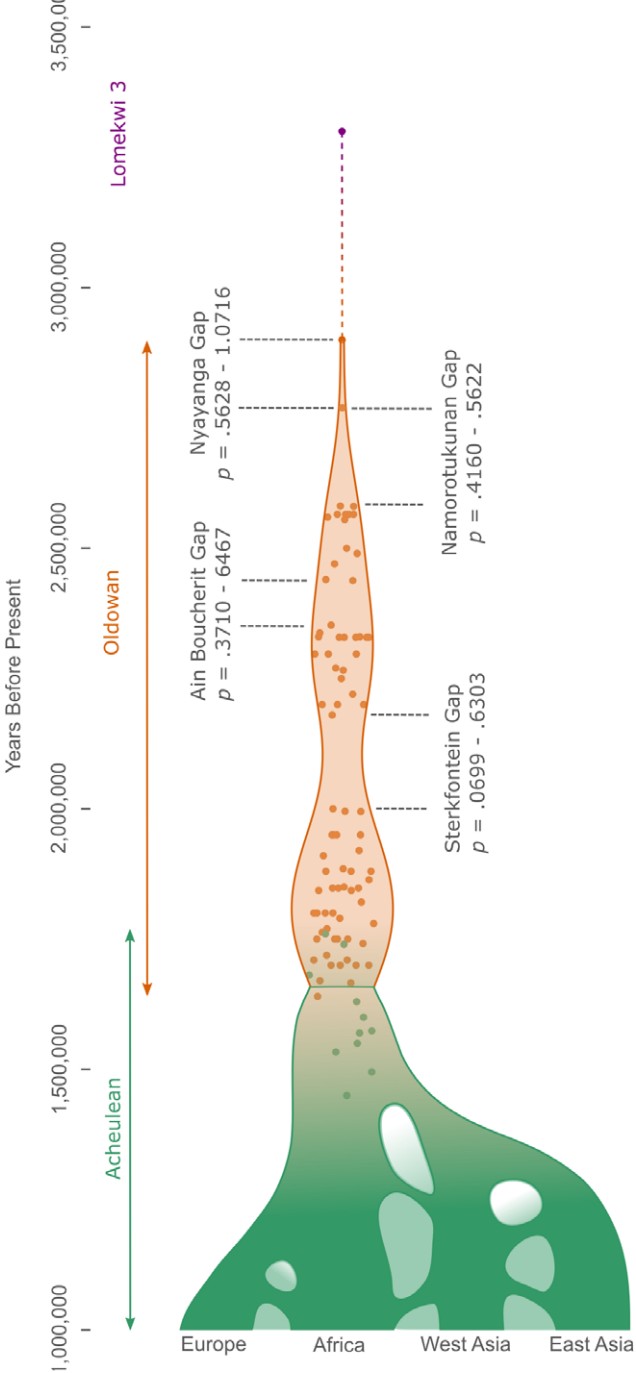

**Figure 1.** The temporal distribution of all known Oldowan sites in Africa (orange), the Lomekwi 3 site (purple) and early Acheulean sites in Africa (green) before its dispersal (e.g., Pappu et al., 2011) into Eurasia. All dated African archaeological sites before 1.5 million years ago are represented. All temporal gaps are highlighted, along with their associated results.

First, the test was applied to the central age estimates. As all sites were required to represent independent cultural occurrences, if age-range overlap was identified or central ages were identical when age ranges did not exist, in sites located <10 km from each other (given Oldowan raw material transportation distances; Braun et al., 2008), all bar one was excluded, with preference for inclusion given to the earliest data point (Table 1). All sites in Table 1 contributed to one or more central-age models.

**Table 1.** The 25 Oldowan occurrences used in the analyses, their temporal data and references for where these data were procured. 'Test rank' refers to the age ranking of sites after those with localised (<10 km) date-range overlap were removed, while 'site rank' refers to a site's age ranking within the complete sample of 91 Oldowan occurrences. As the only site not described by Williams et al. (2025), it is worth highlighting that three Namorotukunan layers are described by Braun et al. (2025) but only two are included in the analyses. As the younger two layer's date ranges are identical and there is potential for their ~2 m vertical separation to have accumulated more quickly than the assumed 140,000 years, and consistent with the sampling procedure described here, only the earlier of these two is included in the modelling.

| Test rank | Site rank | Oldowan site | Central date BP | Date range BP | References |
|---|---|---|---|---|---|
| 1 | 1 | Nyayanga NY–1 | 2,900,000 | 3,032,000–2,595,000 | Plummer et al., 2023 |
| | | Nyayanga Gap (c. 125,000 years) | | | |
| 2 | 2 | Namorotukunan–1 | 2,750,000 | 3,440,000–2,610,000 | Braun et al., 2025 |
| | | Namorotukunan–1 Gap (c. 170,000 years) | | | |
| 3 | 3 | Ledi-Geraru BD 1 | 2,581,000 | 2,610,000–2,581,000 | Braun et al., 2019 |
| 4 | 4 | Namorotukunan–2 | 2,580,000 | 2,610,000–2,137,000 | Braun et al., 2025 |
| 5 | 5 | Gona OGS–7 | 2,565,000 | 2,580,000–2,550,000 | Semaw et al., 2003 |
| 6 | 15 | Ain Boucherit AB-Lw | 2,440,000 | 2,580,000–2,300,000 | Sahnouni et al., 2018 |
| | | Ain Boucherit Gap (c. 90,000 years) | | | |
| 7 | 16 | Hadar A.L. 666 | 2,352,500 | 2,360,000–2,330,000 | Kimbel et al., 1996; Goldman-Neuman and Hovers, 2012 |
| 8 | 18 | Lokalalei LA1A | 2,330,000 | 2,390,000–2,290,000 | Tiercelin et al., 2010 |
| 9 | 20 | Omo Shungura FtJi 1–3–4 Complex | 2,329,000 | 2,334,000–2,324,000 | McDougall and Brown, 2008 |
| 10 | 25 | Omo Shungura 1/E–2 | 2,297,000 | 2,324,000–2,270,000 | Maurin et al., 2017 |
| 11 | 28 | Nasura NAS2 | 2,270,000 | na | Boës et al., 2024 |
| 12 | 29 | Lokalalei 2C (+2A, 2D) | 2,266,000 | 2,390,000–2,266,000 | Delagnes and Roche, 2005; Tiercelin et al., 2010 |
| 13 | 30 | Nasura NAS3 | 2,250,000 | na | Boës et al., 2024 |
| 14 | 31 | Swartkrans Member 1 | 2,220,000 | 2,310,000–2,130,000 | Kuman et al., 2021 |
| 15 | 32 | Gona DAN–2 | 2,200,000 | 2,400,000–2,000,000 | Stout et al., 2005; Domínguez-Rodrigo et al., 2005 |
| 16 | 35 | Sterkfontein Member 5 | 2,180,000 | 2,390,000–1,970,000 | Granger et al., 2015 |
| | | Sterkfontein Gap (c. 180,000 years) | | | |
| 17 | 36 | Kanjera South (Beds KS1–3) | 2,000,000 | 2,300,000–1,920,000 | Ditchfield et al., 2019 |
| 18 | 37 | Olduvai Gorge L.68 | 1,995,500 | 2,015,000–1,976,000 | Stollhofen et al., 2021 |
| 19 | 38 | Drimolen Main Quarry | 1,995,000 | 2,040,000–1,950,000 | Stammers et al., 2018 |
| 20 | 39 | Gona OGS–3 | 1,950,000 | 2,200,000–1,700,000 | Rogers et al., 2023 |
| 21 | 41 | Ileret FwJj 20 (A 41) | 1,950,000 | na | Braun et al., 2010 |
| 22 | 42 | Fejej FJ–1a | 1,950,000 | 1,980,000–1,920,000 | Barsky et al., 2011 |
| 23 | 43 | Ain Boucherit Ab-Up | 1,920,000 | 1,970,000–1,850,000 | Sahnouni et al., 2018 |
| 24 | 44 | Konso KGA6-A1 | 1,910,000 | 1,940,000–1,880,000 | Beyene et al., 2013 |
| 25 | 45 | Kromdraai B | 1,885,000 | 2,000,000–1,770,000 | Braga and Thackerary 2016 |
| 26 | 46 | Koobi Fora FxJj1 105 | 1,880,000 | 1,900,000–1,640,000 | Isaac et al., 1997 |
| 27 | 49 | Olduvai Gorge DK | 1,864,000 | 1,880,000–1,848,000 | Proffitt, 2018 |

Given the temporal uncertainty associated with Oldowan occurrences, we also applied Roberts et al.'s (2023) resampling approach. We drew dates from a normal distribution within each site's age range, where the standard deviation equalled half the difference between the central estimate and the range bounds, and then applied the surprise tests to these randomly generated datasets, repeating the procedure 5,000 times. The mean output of these iterations was used as the resampling result. Neither approach explicitly accounts for the date ranges independently attached to

an occurrence's upper or lower date-range limit. For example, Ar/Ar dating central tendencies may define the lower date threshold above an artefact's sedimentary layer, but the Ar/Ar date itself has its own error range. Our resampling approach and the widespread use of paleomagnetism dating does, however, minimise the impact of this additional error range consideration on our results (Supplementary Information).

For both test versions, a $k$ of 5 and 10 was used following Solow and Smith (2005). The Sterkfontein temporal gap analyses were run

in both forward and reverse directions (Key, 2024). Forward models were not possible for the Nyananga and Namorotukunan temporal gaps, while only a $k = 5$ forward model was possible for the Ain Boucherit gap. Age-range data do not exist for some sites, meaning they could not equally be used during the resampling procedure. In these instances, $k$ was maintained by using the central value date in place of the upper and lower ranges (i.e., effectively creating a resampling range of zero years). Analyses were undertaken in R version 4.3.2 using code available in Roberts et al. (2023).

## Results

No significant results were returned across all models when $\alpha = .05$, with $p \geq .0699$ in all instances (Figure 1a and Table 2). Therefore, none of the investigated temporal gaps were large enough, relative to the temporal spacing of the Oldowan occurrences that preceded or followed them, to infer a loss of cultural information. The null hypothesis that all occurrences were produced by the same cultural process is accepted. Given distributions and cohesion in the rest of the record (Figure 1; Supplementary Information), there is no temporal evidence for a loss of stone tool making knowledge by Oldowan hominins.

## Discussion

These data reveal no temporal evidence for a loss of stone tool making knowledge during the Oldowan. Early *Homo*, *Paranthropus* and potentially *Australopithecus* (Finestone, 2025; Williams et al., 2025) appear to have maintained Oldowan technology as a continuous lineage of cultural information, passed on through generations over an exceedingly long period. Not all species necessarily made lithic tools at all times, and this finding does not

**Table 2.** Significance values using Solow and Smith's (2005) surprise test when applied to the four temporal breaks visible in the Oldowan archaeological ($\alpha = .05$).

| | Model version | k | Reverse p-value | Forward p-value |
|---|---|---|---|---|
| Nyayanga Gap | Central | 5 | 0.5794 | – |
| | Resamp. | 5 | 0.9346 | – |
| | Central | 10 | 0.5628 | – |
| | Resamp. | 10 | 1.0716 | – |
| Namorotukunan Gap | Central | 5 | 0.5306 | – |
| | Resamp. | 5 | 0.5622 | – |
| | Central | 10 | 0.4160 | – |
| | Resamp. | 10 | 0.4406 | – |
| Ain Boucherit Gap | Central | 5 | 0.3710 | .7447 |
| | Resamp. | 5 | 0.3833 | .8750 |
| | Central | 10 | 0.4920 | – |
| | Resamp. | 10 | 0.6467 | – |
| Sterkfontein Gap | Central | 5 | 0.0699 | .1526 |
| | Resamp. | 5 | 0.5662 | .5048 |
| | Central | 10 | 0.1461 | .2264 |
| | Resamp. | 10 | 0.6303 | .6974 |

preclude cultural extirpation events, or non-stone-tool-making populations convergently emulating naturaliths (Eren et al., 2025) or inventing flake tools (Tennie et al., 2017). What it means is that subsequent to the emergence of Oldowan technologies c. 3.0–3.3 Ma (Plummer et al., 2023; Key and Proffitt, 2024), the cultural information linked to this initial event appears to have been maintained ('copied' [c.f., Stout et al., 2019]) as single tradition – if in variable forms and through a braided lineage, potentially with some dead ends – until bifacial core technological components emerged at c. 1.8 Ma (Lepre et al., 2011; Beyene et al., 2013).

Spatial mapping of Oldowan occurrences before and after the temporal gaps does not suggest a shift in the technology's presence (Figure 2), further supporting the case for cultural persistence. Plummer et al. (2023) stress that Nyayanga expands the early Oldowan's spatial range by 1,300 km to southern Kenya, but as a single site separated by a distance easily overcome by population dispersals across c. 125,000 years (especially as Nyayanga, Ledi-Geraru and later Namorotukunan layers feature similar mosaic, $C_4$-dominated environments [DiMaggio et al., 2015; Plummer et al., 2023; Braun et al., 2025]), it is impossible to securely infer a spatial shift. Sites before and after the other temporal gaps are present in both eastern and southern Africa, with no clear differences in their spatial presence (Figure 2).

Technological variation exists in the Oldowan record (Roche et al., 2018), but, at present, there are no marked shifts across these temporal gaps (Braun et al., 2019; Braun et al., 2025; Finestone, 2025). Nyayanga is technologically 'similar to other Oldowan assemblages' (Plummer et al., 2023: 563), including many of the early sites sampled here. Similarly, Namorotukunan 'align[s] with the known Oldowan'; albeit more closely with earlier occurrences (Braun et al., 2025: 9). Across the temporal span of the Oldowan, technological outliers exist, bucking the expected trend of increasing complexity through time. Highly capable flaking is evidenced at 2.3 Ma at Lokalalei 2C (Delagnes and Roche, 2005) for example, while OGS-7 (2.56 Ma), which groups with Braun et al.'s (2019) more complex 'late Oldowan' occurrences (c. 1.6–1.7 Ma), also exhibits more complexity than expected (Semaw et al., 2003; Stout et al., 2010). However, those Oldowan sites before and after the Ain Boucherit and Sterkfontein temporal gaps can be considered similar. Technological and spatial data are therefore consistent with a continuous Oldowan record.

By failing to reject the null hypothesis, our results suggest the processes underpinning prolonged, widespread Oldowan stone tool production – most likely social learning mechanisms (Stout et al., 2019; Sterelny and Hiscock, 2024) – were continuously present across its artefactually evidenced 1.3 million years (Figure 1). That is, the social transmission of Oldowan cultural information appears to have proceeded uninterrupted through generations over an exceptionally long period (c.f., Lycett, 2013). Primate models, experimental data and artefactual (technological) similarities provide a robust foundation for such reasoning (Caruana et al., 2013; Morgan et al., 2015; Stout and Hecht, 2017; Stout et al., 2019; Koops et al., 2022; Wilson et al., 2023; Sterelny and Hiscock, 2024; Braun et al., 2025). We can now add temporal data to the roster of evidence supporting the presence of a single, variable Oldowan cultural lineage across this extended period. Flake stone tools were, therefore, continuously valuable to populations (Shea, 2025) and provided pressure enough for some – be it whole populations or relatively few individuals – to have always found it beneficial to spend time and energy learning the technique. Future modelling/simulation efforts (e.g., Reeves et al., 2023; Cortell-Nicolau et al., 2025) could provide valuable information on the robustness of

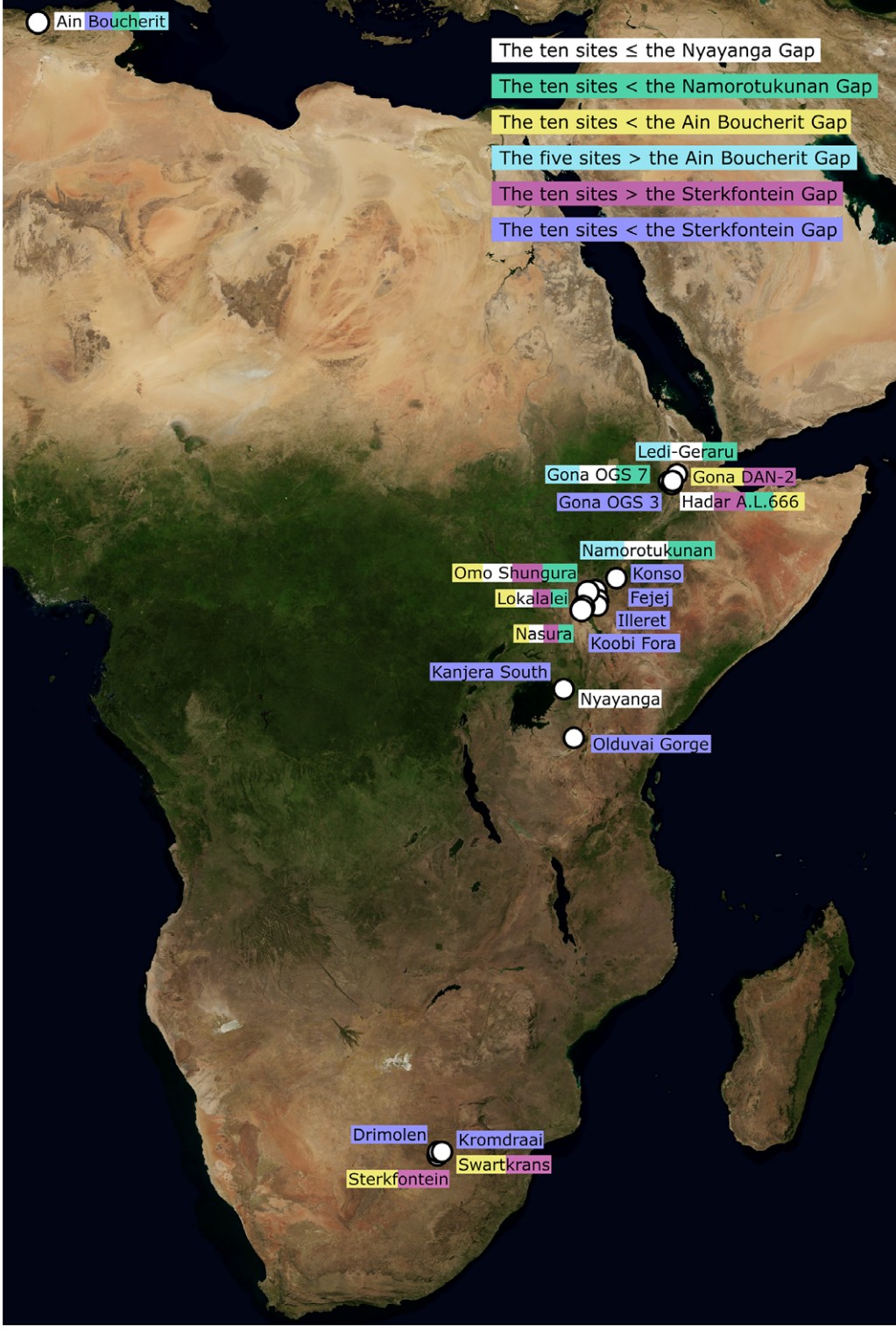

**Figure 2.** A map depicting the Oldowan sites included in the models and the temporal gaps they are used to investigate. No clear spatial differences exist between sites dating before and after the 2.0–2.18 Ma temporal gap. Original satellite image credit: NASA Visible Earth Project.

Oldowan social learning processes in the face of changing ecologies, population pressures and landscape variability.

Our results could also support the continuous presence of alternative mechanisms that explain the Oldowan's persistence. Indeed, all our results do is demonstrate that whatever process was responsible for creating Oldowan artefacts, it would likely have been continually present throughout the period. Social learning processes are the most likely explanation, hence our use of the term 'culture' (Mesoudi, 2016), but it does not exclude the potential for other mechanisms, including the routine loss of technical knowledge followed by its independent reinvention (Tennie et al., 2017), to have created the Oldowan archaeological record. This is an

important theoretical clarification less relevant for later stone technologies, which are (near) universally considered to have been socially maintained traditions (e.g., Lycett, 2013; Lycett et al., 2016; Shipton, 2019; Wilkins, 2020; Key, 2022).

The four investigated temporal gaps may wholly or partially be derived from dating technique limitations, meaning some assemblages were feasibly produced during the investigated gaps, further strengthening evidence of temporal cohesion. If present temporal data do meaningfully reflect Oldowan cultural dynamics, fewer sites may be evidence of smaller tool-producing populations (Figure 1). Evidence of tool-making continuity over c. 300,000 years at Namorotukunan (Ethiopia) – an Oldowan site with marked environment change across its artefact layers – supports our finding of ESA cultural robustness (Braun et al., 2025). Our results tally with Flicker and Key's (2023) finding that, from a temporal perspective, the 3.3 Ma Lomekwi 3 (Kenya) stone tool occurrence should '*currently be considered part of the same cultural process* (i.e.*, not to result from technological convergence*)' as the Oldowan. Key (2022) similarly revealed the early Acheulean record of Africa to be temporally cohesive.

Combined with these prior studies, the present data evidence a continuous record of early human stone tool production in Africa from c. 3.3 to 1.5 Ma. ESA hominin adaptive strategies, therefore, appear to have continuously placed value on the use of stone tools (Shea, 2025). This value would have varied in time and space, but the costs (e.g., Torrence, 1989; Caruana, 2020) of producing and using these tools never wholly outweighed their benefits. Moreover, any influence exerted by stone tool production and use on hominin cognitive and anatomical evolution could have been ever present. All of these inferences are balanced against the coarseness of the archaeological record, but until additional site discoveries or dating method improvements suggest otherwise, the best-fit scenario for the ESA is one of cultural persistence.

**Open peer review.** To view the open peer review materials for this article, please visit http://doi.org/10.1017/ext.2025.10009.

**Supplementary material.** The supplementary material for this article can be found at http://doi.org/10.1017/ext.2025.10009.

**Data availability statement.** All required data for re-running the analyses are available here or in Williams et al. (2025). The relevant code is freely available in Roberts et al. (2023).

**Acknowledgements.** The authors are grateful to the co-authors of the Williams et al. (2025) article for assistance during the construction of the database. EW was supported by an ERC Advanced Grant (BICAEHFID, Grant Agreement: 832980) when the database was constructed.

**Author contribution.** AK conceived the research and undertook the analyses. EW collated the database. Both authors wrote the manuscript.

**Financial support.** This work received no financial support.

**Competing interests.** The authors declare none.

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
