## [Reviewer Report]

The authors use a relatively straightforward method that they have used before in another context to show that two long apparent gaps in the temporal distribution of Oldowan assemblages from eastern and southern Africa are not surprising in the light of the known distribution of temporal gaps and should not be considered as evidence of a break in the Oldowan tradition. This seems convincing.

In the discussion they say, 'Not all species 149 necessarily made lithic tools at all times, and this finding does not preclude cultural extirpation 150 events, or non-stone-tool-making populations convergently emulating naturaliths (Eren et al., 151 2025) or inventing flake tools (Tennie et al., 2017). What it means is that subsequent to the

152 emergence of Oldowan technologies c. 3.0 - 3.3 Ma (Plummer et al., 2023; Key and Proffitt, 2024) 153 the cultural information linked to this initial event appears to have been maintained (‘copied’ [c.f., 154 Stout et al., 2019]) as single tradition – if in variable forms and through a braided lineage,'.

In the light of these relevant points, I would have liked to see more discussion of how the transmission and selection processes would have worked, given that we’re talking about ~700 generations and that populations must have been small. Maybe some suggestions for future simulation work to explore this?

155 potentially with some dead-ends – until bifacial core technological components emerge c.1.8 Ma

156 (Lepre et al., 2011; Beyene et al., 2013).

---

## [Reviewer Report]

The authors of this brief paper aim to address persistent questions about whether the similarity among Oldowan assemblages spanning >1 my represents continuity in cultural transmission, or whether there was loss and reinvention of “baseline” cultural behaviors relating to flaking stone tools. To do this, they analyze dates for all known Oldowan occurrences from East and South Africa, to look for gaps that might indicate that hominins had stopped making stone tools. Their main conclusion is that there is no evidence for any major gaps in dates of Oldowan assemblages. A statistical analysis (Solow and Smith’s “surprise test”) shows that one visually apparent gap is (2.0-2.18 my) is not actually beyond the bounds of expectation for random gaps in the age record, so does not necessarily represent evidence for a long period of discontinuity.

The authors’ conclusions are summed up in the following passage:

“These data reveal no temporal evidence for a loss of stone tool making knowledge during the Oldowan…. What it means is that subsequent to the emergence of Oldowan technologies c. 3.0 - 3.3 Ma (Plummer et al., 2023; Key and Proffitt, 2024) the cultural information linked to this initial event appears to have been maintained (‘copied’ [c.f., Stout et al., 2019]) as single tradition – if in variable forms and through a braided lineage, potentially with some dead-ends – until bifacial core technological components emerge c.1.8 Ma (Lepre et al., 2011; Beyene et al., 2013).”

There is a lot to like about this paper. The authors have made a concerted attempt to use available chronological information to test an interesting and potentially important hypothesis. As far as I can tell, they have been judicious and fair in deciding which cases to include or exclude. The methods applied seem appropriate. And their conclusions are plausible,

One minor question. For the analysis, the authors selected “…dates from a normal distribution within each site’s age-range, where the standard deviation equaled half the difference between the central estimate and the range bounds…”. In some of the cases, the sediments containing the archaeological materials are directly dated, but in most instances they are sandwiched between two dated layers (usually tephras). How did the authors deal with uncertainties of the age estimates from over- and underlying dated deposits? This is not a make-or-break issue. Including the extra uncertainties would almost certainly make it even harder to confirm significant gaps in the record.

While I appreciate the authors’ approach, I do not think that the statistical analysis actually justifies their conclusions. As they state “The method tests the null hypothesis that the outlying record “was generated by the same process” that created the earlier or later records (Roberts et al., 2023: 464). Simply, is the outlying record exceptional relative to the temporal distribution observed in the larger sample of values?”

The unanswered question is, what is this “same process”? From their concluding statement, the authors assume that the process was cultural transmission of some sort. However, their results cannot actually exclude the alternative hypothesis. The “same process” could be what Tennie and colleagues have proposed, repeated, short-term, loss (or abandonment) of technological knowledge, followed by re-invention, guided by fracture mechanics and environmental inheritance. The analyses can rule out temporal gaps on the order of 100’s of thousands of years, but shorter gaps, of a generation or two, or 50 for that matter, would be largely invisible given, the resolution of the dating.

In sum, this paper contains a useful and rigorous analysis of available chronological information about Oldowan sites in East and South Africa. It makes a strong case that there are no major gaps in the age distribution that might indicate rare and prolonged loss of tool making culture. Their conclusion that “… the cultural information linked to this initial event appears to have been maintained (‘copied’ [c.f., Stout et al., 2019]) as single tradition…)” remains plausible. On the other hand, the evidence and methods cannot exclude the alternative hypothesis, that loss and reinvention of technological knowledge were frequent and regular events, below the detection threshold of the available coarse-grained chronological information.

---

## [Editor Report]

Both reviewers are positive about the manuscript and have suggestions for minor revisions.  Reviewer 1 requests greater discussion of how cultural transmission would have operated in this system.  Reviewer 2 asks an important question about whether the model used to account for dating error is appropriate given the nature of age determination for Oldowan sites.  A second comment pertains to question of whether shorter time gaps could be present but undetectable.

---

## [Editor Report]

Thank you for your careful consideration of the editors and reviewers comments. I am happy to accept the revised manuscript.